# Contrasting patterns of *Asaia* association with *Plasmodium falciparum* between field-collected *Anopheles gambiae* and *Anopheles coluzzii* from Cameroon

Claudine Grâce Tatsinkou Maffo,[1,2] Maurice Marcel Sandeu,[1,3] Micareme Tchoupo,[1] Fleuriane Metissa Dondji Kamga,[1] Leon M. J. Mugenzi,[1] Flobert Njiokou,[1,2] Grant L. Hughes,[4] Charles S. Wondji[1,5]

**ABSTRACT**  The widespread prevalence of *Asaia* in mosquitoes makes it a potential candidate for paratrangenic control in *Anopheles*. To better understand whether this bacterium could be used for malaria control, we quantified *Asaia* in *An. gambiae s.l* populations in malaria endemic regions examining co-infection with *Plasmodium falciparum*. Adult *Anopheles* mosquitoes were collected across two different eco-geographical localities in Cameroon, during both the dry and wet seasons. DNA was extracted from whole individual mosquitoes, and real time-qPCR amplification of the *16S ribosomal* RNA was used to quantify *Asaia* in both *An. gambiae* and *An. coluzzii* samples. We also detected and quantified *P. falciparum* infection in the same mosquitoes. The density of *Asaia* was successfully quantified in a total of 864 field mosquitoes, comprising of 439 *An. gambiae* from Bankeng and 424 *An. coluzii* collected from Gounougou. Interestingly, a higher prevalence of *Asaia* in *An. gambiae* (88.3%) compared to *An. coluzzii* (80.9%) was observed. Moreover, the density of *Asaia* in both species was significantly affected by seasonal changes in the two localities. Furthermore, a significant difference between the infection densities of *Asaia* and the *Plasmodium* infection status in the two species was recorded. However, no correlation was observed between the number of *Asaia* and *P. falciparum* infections. This study provides evidence that naturally occurring *Asaia* infection is not correlated to *P. falciparum* development within *An. gambiae* and *An. coluzzii*. Nevertheless, further studies incorporating experimental infections are required to better investigate the correlation between *Anopheles* mosquitoes, *Asaia,* and *Plasmodium*.

**IMPORTANCE**  The symbiont *Asaia* has emerged as a promising candidate for paratransgenic control of malaria, but further analysis of its biology and genetics across Africa is necessary. In this study, we investigated and quantified the influence of *Asaia* in naturally infected *An. gambiae s.l.* populations with the malaria parasite *Plasmodium falciparum*. Genomic DNA was extracted from whole individual mosquitoes collected from two localities, and *Asaia* was quantified using real-time qPCR by amplification of the 16S ribosomal RNA gene. We also detected and quantified *Plasmodium falciparum* infection in the same mosquitoes and established the correlation between *Asaia* and *Plasmodium* coinfection. This study provides evidence that naturally occurring *Asaia* infection is not correlated with *P. falciparum* development within *An. gambiae* and *An. coluzzii* mosquitoes.

**KEYWORDS**  *Asaia, Anopheles gambiae, Anopheles coluzzii, Plasmodium falciparum*, seasons

Address correspondence to Maurice Marcel Sandeu, Marcel.Sandeu@crid-cam.net.

Claudine Grâce Tatsinkou Maffo and Maurice Marcel Sandeu contributed equally to this article. The co-first authors share equal credit for their contributions to the work.

The authors declare no conflict of interest.

See the funding table on p. 12.

Despite the significant progress achieved over the past decades, malaria remains the world's deadliest infectious disease with over 200 million reported cases each year (1). According to the latest estimates from the World Health Organization, 249 million cases of malaria and approximately 608,000 deaths were recorded in 2022. African shoulders the heaviest malaria burden with 233 million malaria cases and 580, 000 deaths, with the majority of deaths occurring in children under 5 years of age (2). The transmission of malaria-causing parasites to humans is exclusively achieved by *Anopheles* mosquitoes, of which five species (*An. gambiae* , *An. funestus*, *An. arabiensis*, *An. moucheti*, and *An. nili*) have been identified as the major malaria vectors in Africa (3). The use of insecticides against vector mosquitoes and antimalarial drugs remains the mainstay of current malaria control. Unfortunately, studies have shown that the effectiveness of these measures is threatened by the rapid increase in insecticide resistance in vector populations (4–9). Clearly, additional strategies are needed in order to combat this deadly disease. The possibility to develop new mosquito control strategies that either suppress population size (reduction or elimination of the wild mosquito population) or replace the population with mosquitoes that are unable to transmit pathogens provides a promising alternative vector control strategy.

The use of microbial symbionts to reduce the competence of vectors involved in pathogen transmission has gained much attention as an emerging alternative approach toward disease control (10, 11). Paratransgenesis exploiting symbiotic bacteria has been proposed as a tool in fighting malaria, mainly by killing *Plasmodium* ookinete/oocysts in the mosquito midgut (12, 13). Several symbiotic bacterial species isolated from mosquitoes have been explored as potential paratransgenic candidates for malaria vector control. For example, *Serratia marcescens AS1* can propagate through mosquito populations, while *Pantoea agglomerans* can significantly reduce *P. falciparum* development when engineered to secrete antiplasmodials (14, 15). Also, *Serratia* sp. are major opportunistic pathogens in humans, causing many nosocomial infections every year (16), and certain strains can be pathogenic in some mosquito systems (17).

Previous studies have identified *Asaia* as a promising candidate that colonizes the ovaries, testes, salivary glands, and the midgut of mosquitoes, spreading both horizontally and vertically through mosquito populations (18–21). This bacterium colonizes a number of arthropods, especially those that feed on the plant nectar, including *Anopheles stephensi*, *An. gambiae, An. funestus, An. maculipennis, Aedes aegypti*, *Ae. albopictus*, *Culex pipiens*, and *Scaphoideus titanus*, many of which can transmit human pathogens (18, 21–27). Importantly, bacteria in the genus *Asaia* appear to not be pathogenic to humans (28–31). Moreover, the presence of *Asaia* in various species of *Anopheles* could be beneficial given that several species are vectors. Therefore, the high prevalence of *Asaia* in these *Anopheles* mosquito species offers a compelling opportunity to study how this symbiont bacterium interacts with *Plasmodium* in the mosquito host.

In this study, we investigated the effect of *Asaia* on *P. falciparum* infection in a natural *An. gambiae* s.l population from two different eco-geographical localities in Cameroon. We screened mosquitoes to detect the presence of *Asaia* and *P. falciparum* using a newly developed qPCR assay. We further quantified the density of both microorganisms in *An. gambiae* s.l samples. With this approach, we empirically assessed the effect of *Asaia* on *P. falciparum* infection in mosquitoes under natural conditions. Findings from this study will contribute to our understanding of the tripartite interactions between symbionts, the mosquito host, and *Plasmodium* parasite. Deciphering these complex interactions will also provide more evidence on the use of *Asaia* symbionts as a potential candidate for alternative vector control tool.

## MATERIALS AND METHODS

### Study sites and mosquito sampling

The mosquitoes used in this study originated from field and laboratory-reared colonies of adult *Anopheles* spp. Laboratory-reared mosquitoes were used for bacteria isolation, and field mosquitoes were used to identify and quantify *Asaia* bacteria. Field mosquito sampling was conducted in two localities in Cameroon, namely, Gounougou (northern region 9°03′00″N, 13°43′59″E) and Bankeng (central region, 4°40′26.4″N, 12°22′30″E.). Wild populations of female *An. gambiae* were collected between 06:00 am and 09:00 am in both study sites during the dry season from December 2018 to January 2019 and during the wet season in August 2019. Mosquitoes were collected using Prokopack electrical aspirators (John W. Hook, Gainesville, FL, USA) following verbal consent from the village chief and each household representative. The collected mosquitoes were kept in individual tubes and subsequently transported to the insectary at the Centre for Research in Infectious Diseases (CRID), Yaoundé, for further experiments. In the insectary at the CRID, these mosquitoes have been kept for 7 days post-collection, which is the required time for *Plasmodium* to develop in the *Anopheles* mosquito vectors.

### *Asaia* isolation and morphological identification

Laboratory-reared *Anopheles* mosquitoes (*An. gambiae* s.l.) were knocked down by incubating them for 2 minutes at −20°C, and their surfaces were sterilized by washing them in 70% ethanol and rinsing them in distilled water. The isolation of *Asaia* then proceeded as described previously by Rami *et al.*, 2012 (32) with slight modifications. After sterilization, five pools of five mosquitoes each were suspended and ground in 1 mL of sterile saline solution (0.9% NaCl). Then, a 0.5-mL aliquot of this suspension was inoculated into a specific enrichment broth comprising 2.2% D-sorbitol, 0.5% peptone, 0.5% yeast extract, and 100 ppm cycloheximide with the pH adjusted to 3.5 with hydrochloric acid for the isolation of acetic acid bacteria. Next, isolated bacteria were inoculated in 10.0 mL of the enrichment culture medium in culture tubes. The extent of microbial growth at 30°C was determined by monitoring the turbidity. Subsequently, the microorganism suspension was spread on a $CaCO_3$ agar plate containing 2.2% D-glucose, 1.0% ethanol, 1.0% yeast extract, 0.7% $CaCO_3$, and 1.2% agar. *Asaia* colonies were experimentally identified based on the morphological properties and formation of carbonate dissolution haloes in agar plates. All the colonies that could clear the $CaCO_3$ were selected and isolated for further experiments.

### DNA extraction from bacterial isolates, field mosquitoes, and mosquito species identification

Prior to genomic DNA extraction, individual adult female mosquitoes were surface-sterilized by washing them in 70% ethanol for 5 minutes and then rinsing twice with sterile distilled water in order to remove superficial bacteria to avoid external contamination. DNA was extracted from whole individual mosquitoes and bacterial isolates using the Genejet extraction kit following the manufacturer's recommendations. DNA extracts were eluted in a final volume of 100 µL and stored at −20°C for further analysis. Molecular species identification was undertaken using diagnostic species-specific PCR assays to distinguish between the morphologically indistinguishable sibling mosquito species of the *An. gambiae* s.l. complex using the short interspersed elements (SINE) (33).

### qPCR for the detection and quantification of *Asaia* in field mosquitoes

*Asaia* detection targeting the 16S ribosomal RNA gene using primers AsaH1 (F,5′- AAG GGC GCG TAG GCG GTT TA-3′) and AsaR (R, 5′-CCA CCG GTG TTC TTC CCA AT-3′) designed in this study using Primer 3 design tool from the sequence obtained through 16S ribosomal RNA Illumina sequencing (34) was undertaken on a subset of samples collected. qPCRs were prepared using 5 µL of FastStart SYBR Green Master mix, a final

concentration of 1 µM of each primer, 1 µL of PCR-grade water and 2 µL template DNA to a final reaction volume of 10 µL. Prepared reactions were run on a Stratagene MX3005P (Agilent Technologies, Santa Clara, California, USA) System for 15 minutes at 95℃, followed by 40 cycles of 95℃ for 15 seconds and 58℃ for 30 seconds. Amplification was followed by a dissociation curve (95℃ for 10 seconds, 65℃ for 60 seconds, and 97℃ for seconds) to ensure the correct target sequence was being amplified. Cycle threshold (CT) values were based on duplicate samples. Positive controls included a previously cultured *Asaia*-infected sample. A standard curve was generated from tenfold nine serial dilutions of the cultured *Asaia* DNA (two technical replications for each dilution) corresponding to a range of $1.44 \times 10^5$ to $1.44 \times 10^{12}$ genome/µL. The bacterial colony copy numbers were calculated using the following formula (35): Copy number= (DNA amount (ng) $\times 6.022 \times 1{,}023$/length (bp) $\times 109 \times 650$). All amplification products were quantified and normalized using the standard range achieved with the *Anopheles* ribosomal gDNA purified PCR product using primers RPS7 (F, 5′- CCA CCA TCG AAC ACA CAA AGT TGA-3′; R,5′- TGC TGC AAA CTT CGG CTA TTC-3′) (36). In order to compare bacterial densities between individual samples, the relative ratio was calculated by dividing the amount of *Asaia* DNA obtained by absolute quantification by the amount of the housekeeping DNA (S7) determined in the same sample.

## Sequencing and phylogenetic analysis of bacterial isolates and PCR-positive samples

To confirm the bacterial isolates, the amplified bacterial isolate products were sequenced by Sanger sequencing. In order to investigate whether the strains of *Asaia* infecting field mosquitoes differed from one season to another in both localities, samples from Bankeng and Gounougou were subjected to PCR as previously described (21) and sequenced.

Overall, four colonies of isolates and 57 individual field samples were analyzed using the 16S ribosomal RNA gene. Among these 57 samples, the distribution included 19 and 20 samples collected from Bankeng during the dry and rainy season, respectively, and five and 13 samples collected from Gounougou during the dry and rainy seasons, respectively. The number of samples used was based on the number of infected individuals in the various species and the sharpness of the bands after the gel electrophoresis.

PCR products of *Asaia*-positive samples were purified using Exonuclease I (Exo I) and Shrimp Alkaline Phosphate (Exo-SAP protocol) according to the protocol of New England Biolabs (NEB, MA, and USA) and sequenced. Maximum likelihood phylogenetic trees were constructed for the *Asaia*-specific 16S ribosomal RNA gene target. The taxonomic relationships of the strains obtained from this study were inferred against GenBank sequence isolates. These sequences were aligned with previous existing isolates using the ClustalW multisequence alignment tool in BioEdit software. *Neoasaia chiangmaiensis* (FJ887939.1), *Gluconobacter oxydans* (KU255083.1), and *Acetobacter tropicalis* (JF930138.1) were used as outgroups. The evolutionary history was inferred using the maximum likelihood method using MEGA X based on the Jukes–Cantor model. The robustness of the individual branches was estimated by bootstrapping with 1,000 replicates.

## Real-time PCR for the detection and quantification of *Plasmodium falciparum* in mosquitoes

The detection of *Plasmodium* infection from each whole individual mosquito was performed using TaqMan assay, as previously described (37). Absolute quantification of all positive specimens was done using the standard curve of the 3D7 strain DNA and generated from serial dilution methods, resulting in a quantification range of $7.10^1$ to $7. 10^5$ genomes/ µL. All amplification products were quantified and normalized using the standard range achieved with the S7 gDNA purified PCR product. In order to compare parasite densities between individual samples, the relative ratio was calculated

by dividing the amount of *P. falciparum* DNA obtained by absolute quantification by the amount of the housekeeping DNA (S7) determined in the same sample.

## Statistical analysis

Statistical analysis was performed in GraphPad Prism 7.05, and alpha values for significance were taken at $P < 0.05$, with all confidence intervals (CIs) at 95%. The prevalence of *Asaia* and *Plasmodium* was compared using the χ test, while intensities of infection were compared using the nonparametric Mann–Whitney test. The correlation between *Asaia* and *Plasmodium* amounts has been measured using Pearson correlation test.

## RESULTS

### Morphological and molecular characterization of *Asaia* isolates

Four types of colonies were obtained after culturing, distinguished by their differences in colors (Fig. 1A). PCR analysis confirmed that each isolate identified belongs to the *Asaia* genus. The constructed phylogenetic tree showed that these isolates were identical, clustered together, and fall into a same clade as previously identified *Asaia* strains Fig. 1B). These sequences obtained in this work were submitted to the GenBank database with the accession numbers ON54736–ON54755.

### Mosquito identification and *Plasmodium* spp. infection rate

Overall, 1,985 mosquitoes were collected, including 975 from Bankeng and 1,010 from Gounougou. A total of 425 were collected during the dry season and 1,560 in the rainy season. Molecular identification was conducted on all the 425 mosquitoes collected

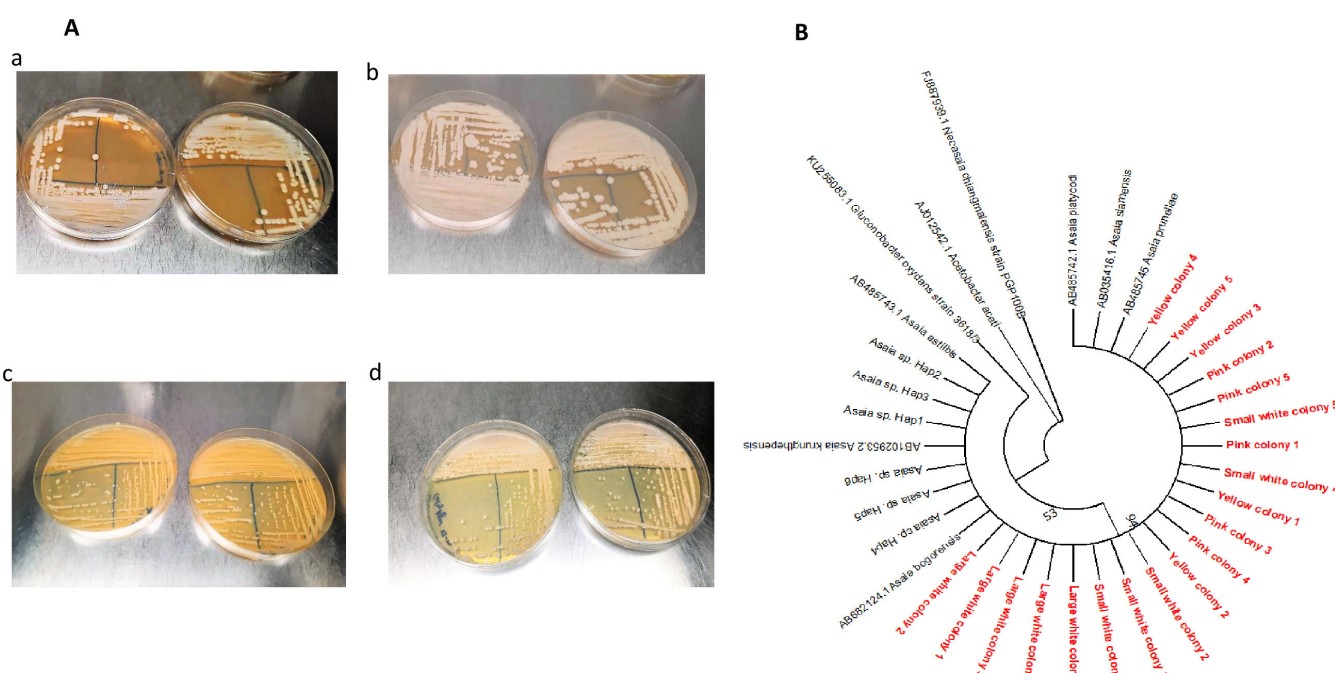

**FIG 1** (A) The appearance of *Asaia* colonies on the glucose–yeast extract–CaCO3 medium. Four types of colonies were obtained after culturing, with the colony ranging in colors from pink to yellowish, white, and smooth. *Asaia* species produce white medium noninvasive (a), white large invasive (b), yellow smooth (c), and bright pink (d) colonies on the glucose–yease extract–CaCO3 medium, which is one of their unique morphological characteristics. (B) Evolutionary analysis of the isolated *Asaia* bacteria. A phylogenetic tree was created with MEGAX software and ClustalW program based on 16S rRNA gene sequences. The maximum likelihood method was used to infer evolutionary relationships, and associated taxa were clustered together in the bootstrap test (1,000 replicates). The red-colored branches represent the sequences of the different isolated *Asaia* colonies according to their appearnce, while the black branches represent the previously described sequences of *Asaia*.

during the dry season and 669 (335 and 334 for Bankeng and Gounougou, respectively) randomly selected mosquitoes from the rainy season. *An. gambiae* was predominant (*n* = 555, 99%) at Bankeng, during both seasons, with only three *An. coluzzii* mosquitoes. On the contrary, mosquitoes from Gounougou were predominantly *An. coluzzii* (*n* = 503, 93%) and a smaller proportion of *An. arabiensis* 7% (*n* = 36). A subset of 1,058 of mosquitoes from Bankeng (*n* = 555) and Gounougou (*n* = 503) was used to assess *Plasmodium* infection using a TaqMan assay. The overall prevalence of *P. falciparum* was 16.26%. Specifically, we found a *P. falciparum* prevalence of 20.1% (112/555) at Bankeng for *An. gambiae* and 11.9% (60/503) at Gounougou for *An. coluzzii,* indicating a higher prevalence of *P. falciparum* in *An. gambiae* than in *An. coluzzii.*

## Prevalence, intensity of *Asaia* infection, and strain diversity according to the season

A subset of 864 mosquitoes was tested by qPCR for *Asaia,* including 439 *An. gambiae* and 424 *An. coluzii* samples from Bankeng and Gounougou, respectively. In Bankeng, where only *An. gambiae* was identified, the prevalence of *Asaia* was 88.3%, while it was 80.9% among *An. coluzzii* mosquitoes in Gounougou (Table 1). A comparison of the medians of infection densities showed that *An. gambiae* mosquitoes from Bankeng had a significantly higher bacterial load than *An. coluzzii* mosquitoes from Gounougou (*P* < 0.0001) (Fig. 2A). Moreover, assessing the impact of seasonal variations on the infection rate of *Asaia* in both species revealed a significant difference in the infection densities of *Asaia* in the two species per season in both localities. Mosquitoes collected in the rainy season (median of 13,590) had a higher bacterial density of *Asaia* than those collected in the dry season (median of 64.49) (*P* < 0.0001) (Fig. 2B). In contrast, bacterial densities were significantly higher in the dry season than in the rainy season (*P* < 0.0001) in Gounougou (Fig. 2C). The prevalence values of *Asaia* according to the seasons and localities are presented in Table 2.

In addition, we investigated the genetic diversity of *Asaia* in *An. gambiae* and *An. coluzzii* according to the seasons. The sequences obtained were subjected to BLAST search in GenBank (NCBI), and all had an identity percentage ranging from 99% to 100% with *Asaia*. The relationship between *Asaia* and *Anopheles* mosquitoes revealed clustering of *Asaia* strains in both localities irrespective of the season (Fig. 3A), suggesting that season does not have an impact on the distribution of *Asaia* strains. By analyzing the haplotype network, we found six distinct nucleotide sequence variants (Fig. 3B), where the major one H2 (39 samples) and the haplotype H1 (nine samples) were present in Bankeng and Gounougou in both seasons. In addition, the nucleotide sequence variants H4 (two samples), H5 (two samples), and H6 were present only in Bankeng with H6 occurring (one sample) only in the rainy season and the others in the dry season. Furthermore, the variant H3 (four samples) is present only in Gounougou and in the rainy season. These results can suggest that the variants are not entirely locality-dependent.

## Correlation between *Asaia* sp. and *Plasmodium falciparum* infections

Beyond the spatiotemporal variation of *Asaia* bacteria in *An. coluzzii* and *An. gambiae*, the correlation of *Asaia* infection and *Plasmodium* infection status was also evaluated. The observed proportion of females coinfected with *Asaia* and *Plasmodium* was 13% (58/388) in Bankeng (*An. gambiae*) and 14% (60/423) in Gounougou (*An. coluzzii*). The prevalence of *Asaia* was 79.45% (58/73) in *Plasmodium*-infected mosquitoes and 90.2% (330/366) in *Plasmodium*-uninfected mosquitoes in Bankeng. In Gounougou, all *Plasmodium*-infected

**TABLE 1** Prevalence of *Asaia* according to localities or mosquito species

| Localities | *Asaia* positive (%) | *Asaia* negative (%) | Total |
|---|---|---|---|
| Bankeng (*An. gambiae*) | 388 (88.38%) | 51 (11.62%) | 439 |
| Gounougou (*An. coluzzii*) | 361 (80.94%) | 63 (19.05%) | 424 |
| Overall | 749 (86.7%) | 115 (13.3%) | 864 |

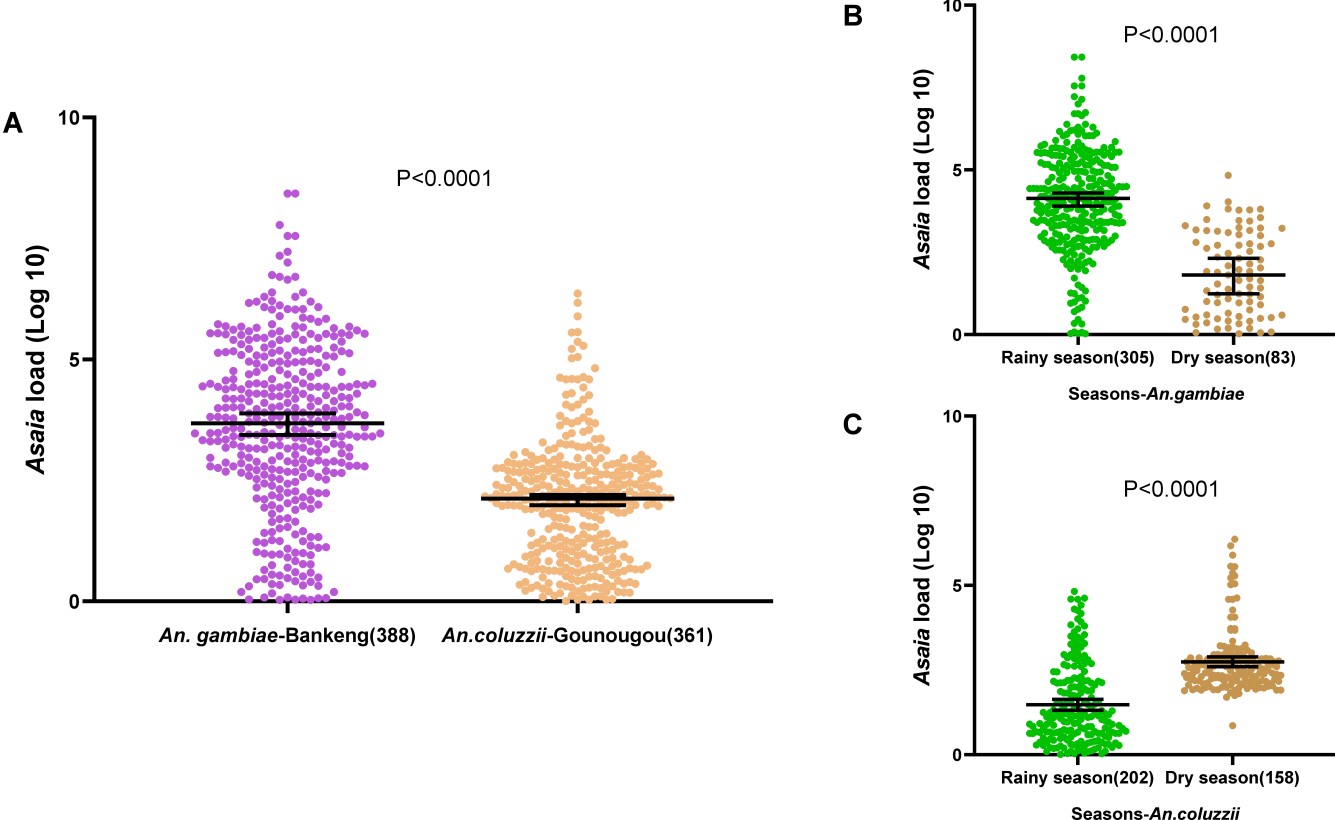

**FIG 2** *Asaia* density in Anopheles mosquitoes. Relative amount of *Asaia* in *An. gambiae* at Bankeng and *An. coulzzii* at Gounougou (A). Relative amount of *Asaia* in *An. gambiae* at Bankeng (B) and *An. coluzzii* at Gounougou (C) during the dry and rainy seasons. The relative amount of *Asaia* was expressed as a ratio of *Asaia* and mosquito rps7 gene copies in log10 values; amounts were median with 95% CI, and they were compared between the both localities/species using the Mann–Whitney nonparametric test. The numbers in the brackets represented the number of samples used for the analysis.

mosquitoes were infected with *Asaia* (100%), while the prevalence of *Asaia* was 82.7% (301/364) in *Plasmodium*-negative mosquitoes (Table 3). The observed proportion of females coinfected with *Asaia* and *Plasmodium* (58/488 = 13%) (Fig. 4A) in Bankeng was similar to the expected proportion (14.1%) based on the prevalence of *Plasmodium* (18.8%) and *Asaia* (75.1%) infection (expected prevalence = 0.188 × .751 = 0.141), indicating that females infected with one organism are likely to be infected with the other. In Gounougou (*An. coluzzii*), 301/423 (71.1%) females were infected with *Asaia*. The observed proportion of females coinfected with *Asaia* and *Plasmodium* (59/423 = 14%) was lower than the expected proportion (9.8%) based on the prevalence of *Plasmodium* (13.9%) and *Asaia* (75.1%) infections (expected prevalence = 0.139 × .711 = 0.0988), indicating that females infected with *Asaia* could facilitate *P. falciparum* infection (Fig. 4B).

The median densities of *Asaia* infection were subsequently compared between *Plasmodium*-infected and noninfected mosquitoes. A significant difference between the infection densities of *Asaia* in *Plasmodium*-infected and -uninfected mosquitoes in the two locations was observed. Specifically, in Bankeng, *Plasmodium*-uninfected *An. gambiae* mosquitoes had a higher density of *Asaia* compared to *Plasmodium*-infected

**TABLE 2** Prevalence of *Asaia* according to localities and seasons

| Localities | Seasons | *Asaia* positive (%) | *Asaia* negative (%) | Total |
|---|---|---|---|---|
| Bankeng | Rainy season | 305 (97.7%) | 7 (2.3%) | 312 |
| | Dry season | 83 (65.3%) | 44 (34.7%) | 127 |
| Gounougou | Rainy season | 202 (76.3%) | 63 (23.7%) | 265 |
| | Dry season | 158 (99.4%) | 1 (0.6%) | 159 |

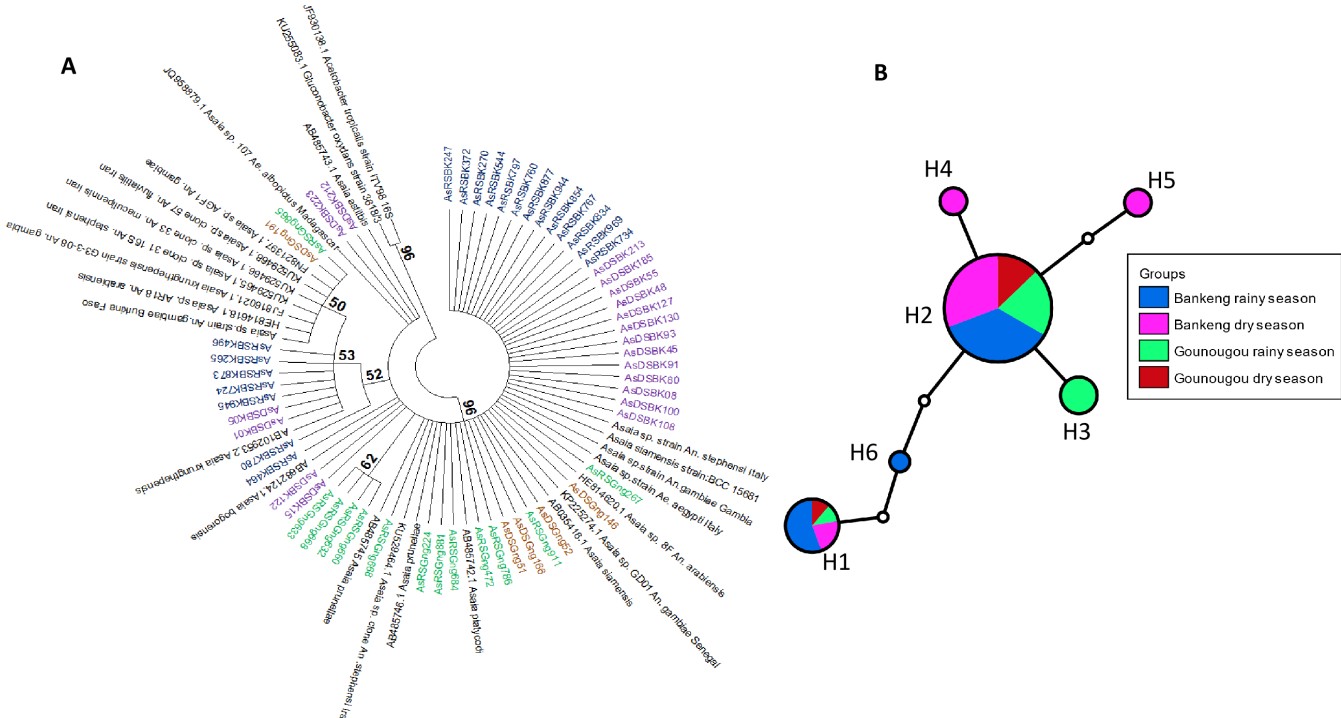

**FIG 3** (A) Phylogenetic tree of the strains of *Asaia* in both seasons in both localities. AsRSBK represent samples of the rainy season at Bankeng (*An. gambiae*) and AsDSBK (purple) samples of the dry season at Bankeng; AsDSGng (blue) and AsRSGng (green) represent samples of the dry and rainy seasons, respectively, at Gounougou (*An. coluzzii*). (B) Nucleotide sequence variant network. Nucleotide sequence variant networks were constructed using *Asaia* 16S sequences in the TCS program. H1–H6 represent the different nucleotide sequence variants, and each color represents each mosquito species according to the locality and season.

mosquitoes ($P < 0.0001$) (Fig. 5A). In contrast, *Plasmodium*-infected *An. coluzzii* had higher *Asaia* loads compared to their *Plasmodium*-free counterparts ($P < 0.0001$) (Fig. 5B).

In order to understand the interaction in more detail, we examined the correlations between *Asaia* and *P. falciparum* densities. Our data showed a very weak positive correlation between *Asaia* and *P. falciparum* ($P > 0.05$) in both localities with similar correlation curves ($R^2 = 0.02$ and $R^2 = 0.05$ for Bankeng and Gounougou, respectively) (Fig. 6A and B). This result reflects an independent evolution of the two organisms, suggesting that *Asaia* and *Plasmodium* do not have an impact on each other in the mosquito populations we studied.

## DISCUSSION

Paratransgenesis is a promising and particularly ingenious strategy currently being developed for controlling vector-borne diseases. It utilizes the genetically manipulated native microbiota of the insect vector to interfere with pathogens within the insect vector. Among the several candidates for paratransgenic control of malaria, *Asaia* is the focus of consideration due to its specific features such as a stable association with *Anopheles* mosquitoes, genetic tractability, and horizontal and vertical transmission

**TABLE 3** Prevalence of *Asaia* according to localities and *Plasmodium* infection status

| Localities | Infection status | *Asaia* positive (%) | *Asaia* negative (%) | Total |
|---|---|---|---|---|
| Bankeng | *Plasmodium+* | 58 (79.45%) | 15 (20.55%) | 73 |
|  | *Plasmodium-* | 330 (90.2%) | 36 (9.8%) | 366 |
| Gounougou | *Plasmodium+* | 60 (100%) | 0 | 60 |
|  | *Plasmodium-* | 301 (82,7%) | 63 (17.3%) | 364 |

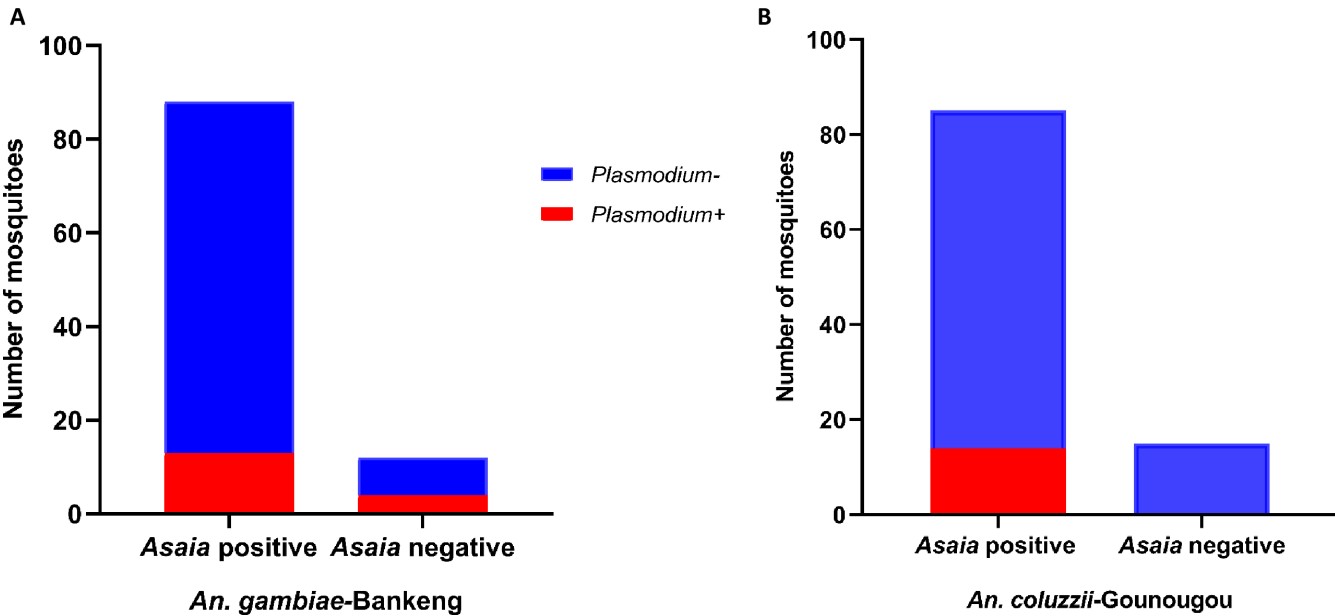

**FIG 4** Prevalence of *Asaia* and *Plasmodium* in *An. gambiae* and *An. coluzzii* mosquitoes from Bankeng and Gounougou, respectively. A) *Plasmodium falciparum* and *Asaia* infection rates (%) in *An. gambiae* females; B) *Plasmodium* and *Asaia* infection rates (%) in *An. coluzzii* females.

modes (21). This study assessed the infection rate and density of *Asaia* and its correlation with *Plasmodium* in *An. gambiae* and *An. coluzzii* mosquitoes in Cameroon.

This study revealed an overall prevalence of *Asaia* in *Anopheles* mosquitoes of 86.7%, with a similar prevalence of *Asaia* between *An. gambiae* and *An. coluzzii,* suggesting that both species are susceptible to the infection by *Asaia* sp. This infection rate of *Asaia* is higher than the one obtained in a previous study conducted in *Anopheles* species in Cameroon (21). This difference in prevalence could be due to seasonal and geographical

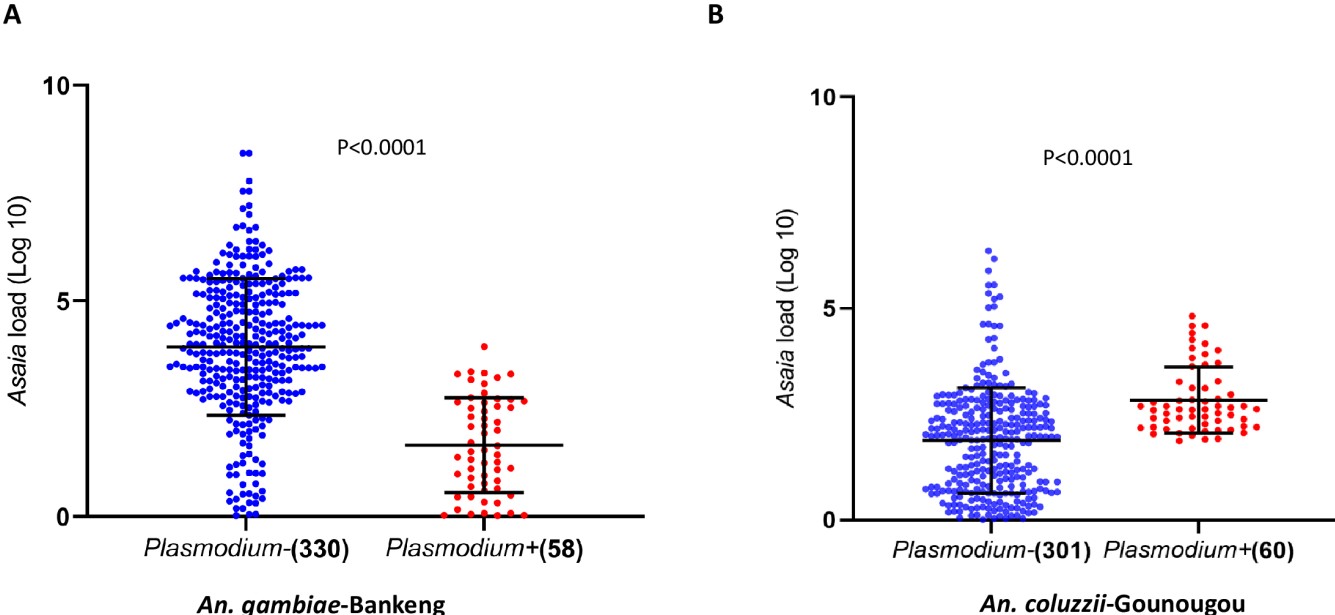

**FIG 5** Relative amount of *Asaia* in *Plasmodium*-infected and noninfected *An. gambiae* (A) *and An. coluzzii* (B). The relative amounts of *Asaia* were expressed as a ratio of bacterial 16S ribosomal RNA and mosquito rps7 gene copies in log10 values; amounts were median with 95% CI and compared between *Plasmo-dium*-infected and noninfected *Plasmodium* individuals using the Mann–Whitney nonparametric test. *P. falciparum* +represents *Plasmodium falciparum*-positive samples and *Plasmodium*-, noninfected *Plasmodium* samples. The numbers in the brackets represent the number of samples used for the analysis.

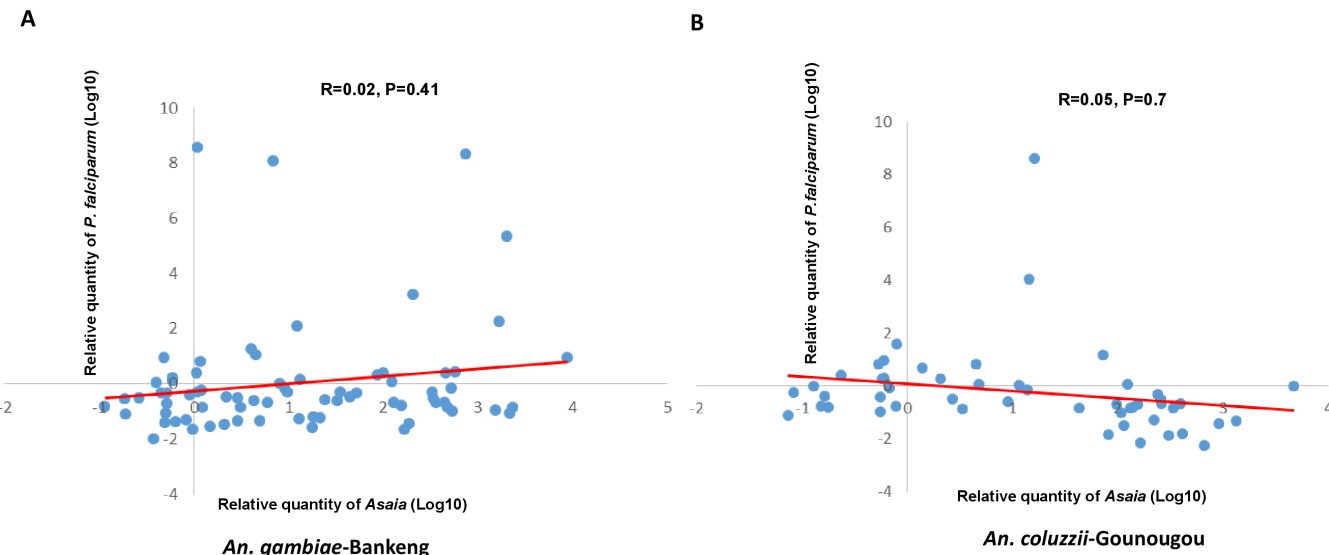

**FIG 6** Correlation curves between *Asaia* and *P. falciparum* densities. The dots represent the relative quantities of *Asaia* obtained from the individual sample presented and the corresponding amount of *Plasmodium* parasites. (A) Bankeng and (B) Gounougou. Pearson's correlation was used to measure the correlation between the amounts of *Asaia* and *Plasmodium* in each locality. R represents the correlation coefficient and P the degree of significance.

heterogeneities in microbiota compositions in *Anopheles* mosquitoes from the various study sites (34, 38, 39). These contrasting results can also suggest that environmental conditions in which mosquito larvae develop are key determinants within the vectors, and this could have important implications on the composition of the microbiota of mosquitoes (34). Also, this difference could be explained by the methods of detection used. In the previous study, the detection was done by conventional PCR, while in this present study, we used quantitative PCR, which is a more sensitive technique (40). We then quantified *Asaia* density in its mosquito host and investigated variations of these densities according to season and locality. We observed that *Asaia* densities varied greatly in all our individual samples. The analysis of the densities of infection showed that *An. gambiae* are more highly infected than *An. coluzzii*. In addition, the results showed a higher density of *Asaia* infection in the dry season compared to the rainy season for *An. coluzzii*; meanwhile, *An. gambiae* mosquitoes collected in the rainy season were more highly infected by *Asaia* than those collected during the dry season. An earlier study demonstrated that the breeding sites of *Anopheles gambiae* s.l. can contain a wide range of microbial communities, and their composition may vary due to interactions between microbes (41). The authors compared the genomes of *An. gambiae* and *An. coluzzii* and found significant differences in their immune genes (41). They hypothesized that divergence is driven by an immune response to microbes specific to the different larval aquatic habitats occupied by these species (41). In addition, this difference in densities of infection can also be due to host age and physiological status, as has been observed in other mosquito species (42, 43). Alternatively, microbe–microbe interactions can influence the prevalence and abundance of microbiome members (44, 45), and *Asaia*, for instance, has been shown to both antagonize and interact with other microbes (27, 46). Since the *Anopheles* mosquitoes we studied were sourced from a natural population, we did not have information regarding their age or physiological stage. This lack of information could account for the observed variations in *Asaia* densities, which in turn could potentially influence *Plasmodium* infection.

*Asaia* has shown promise as a potential candidate for paratransgenic control of malaria due to its ability to negatively affect the development of *Plasmodium* at various stages and its capacity to transmit and persist within field populations of *Anopheles* species (21, 47). In this study, we focused on evaluating the interaction between *Asaia* and *Plasmodium* in natural *Anopheles gambiae* and *coluzzii* populations by quantifying

the amount of *Asaia* present in both *Plasmodium*-infected and noninfected mosquitoes. In *An. gambiae*, *P. falciparum* noninfected mosquitoes had higher densities of *Asaia* compared to *Plasmodium*-infected mosquitoes, suggesting that the presence of *Asaia* may impede the infection by *Plasmodium*. This inhibitory effect on *Plasmodium* parasites has been reported in previous studies (19, 47). In fact, *Asaia* has been shown to act as an immunomodulator within the mosquito population by inducing the production of antimicrobial peptides that may interfere with the course of infection, particularly during the invasion of epithelial tissues and translocation to the salivary glands. *Asaia* may also act to enhance immune genes (19). Alternatively, *Plasmodium* infection may stimulate insect immunity, which subsequently reduced *Asaia* loads.

In contrast, we found that in *An. coluzzii,* the load of *Asaia* was higher in *Plasmodium*-infected individuals than *Plasmodium* noninfected ones. In addition, all *Plasmodium*-positive samples were *Asaia*-infected, suggesting that *Asaia* could facilitate *P. falciparum* infection. However, there is a lack of studies on *Asaia* showing the same pattern, but another study showed that different strains of the bacteria *Serratia* can induce different outcomes on *Plasmodium* infections (48). Bai *et al.* identified two mosquito symbiotic *Serratia* strains *Serratia* Y1 and *Serratia* J1, from field-caught *An. sinensis* mosquitoes, with different contrasting effects on *Plasmodium berghei* development. The *Serratia* Y1 strain showed anti-*Plasmodium* activity, while *Serratia* J1 did not influence parasite development in the midgut of mosquitoes (49). This indicates that the effect of mosquito gut bacteria on parasite infection is complex and may depend on species-specific or strain-specific interactions (48). It was also suggested that, despite the fact that there is an overlay between the mosquito antibacterial and antimalarial immune responses, some immune genes induced by bacteria have a minimal impact on *Plasmodium* development (50–52). It has also been shown that in *Anopheles* mosquitoes, ingestion of a blood meal containing high loads of *Plasmodium* triggers a strong immune response, acting both in the lumen and in the gut wall (53). Indeed, *Plasmodium*-infected blood meals induce an immune response in mosquitoes, which might limit not only the malaria parasite progression in the insect but also the symbiont population. This is the case with *P*. vivax (54), where it has been shown that infectious blood meal leads to dramatic suppression of the gut-bacteria population of mosquitoes. Hence, the midgut microbiota of the *Anopheles* can be negatively affected by this response (19).

In this study, the lower level of *Plasmodium* infection in *An. coluzzii* could be explained by the higher intensity of *Asaia,* suggesting that *Asaia* influence *Plasmodium* development. Indeed, it has been demonstrated that the composition of the gut microbiota may influence parasite transmission, and this may explain the difference in infection levels between mosquito populations from diverse environments (55, 56).

## Conclusion

Here, we isolated and quantified *Asaia* in *An. gambiae* and *An. coluzzii* and characterized the *Asaia–Plasmodium* interactions in these Anopheline species. We showed that seasons and mosquito species impact the abundance of *Asaia* and further, the complexity of the *Asaia–Plasmodium* interaction. However, no correlation was observed between the loads of *Asaia* and *P. falciparum* infection. Further studies applying experimental infection and immunological experiments are therefore needed to better understand the mechanisms involved, in order to better characterize *Anopheles–Asaia–Plasmodium* interactions and the potential for paratransgenic vector control.

## ACKNOWLEDGMENTS

This work was funded by the Medical Research Council, UK, and Global Challenges Research Fund, through the PIIVeC. Grant number MR/P027873/1.

M.M.S. and C.S.W. conceived and designed the study. M.M.S. and C.G.M. collected the samples on the field. C.G.M., M.M.S., and F.M.D. performed the culture and isolation of *Asaia*. C.G.M., M.M.S., and M.T. designed the experiments and performed the validation

of the qPCR assay. C.G.M. performed the data analysis with the help of M.M.S. C.G.M. and M.M.S. wrote the manuscript with contribution from F.N., G.L.H., and C.S.W.

## AUTHOR AFFILIATIONS

[1]Department of Medical Entomology, Centre for Research in Infectious Diseases (CRID), LSTM Research Unit, Yaoundé, Cameroon

[2]Department of Animal Biology and Physiology, Faculty of Science, University of Yaoundé 1, Yaoundé, Cameroon

[3]Department of Microbiology and Infectious Diseases, School of Veterinary Medicine and Sciences, University of Ngaoundéré, Ngaoundéré, Cameroon

[4]Departments of Vector Biology and Tropical Disease Biology, Centre for Neglected Tropical Diseases, Liverpool School of Tropical Medicine, Liverpool, United Kingdom

[5]Department of Vector Biology, Liverpool School of Tropical Medicine, Pembroke Place, Liverpool, United Kingdom

## AUTHOR ORCIDs

Maurice Marcel Sandeu http://orcid.org/0000-0002-6780-5394
Grant L. Hughes http://orcid.org/0000-0002-7567-7185

## FUNDING

| Funder | Grant(s) | Author(s) |
| --- | --- | --- |
| UKRI | Medical Research Council (MRC) | MR/P027873/1 | Maurice Marcel Sandeu |

## AUTHOR CONTRIBUTIONS

Claudine Grâce Tatsinkou Maffo, Data curation, Formal analysis, Methodology, Software, Writing – original draft | Maurice Marcel Sandeu, Conceptualization, Data curation, Formal analysis, Funding acquisition, Investigation, Methodology, Supervision, Validation, Visualization, Writing – original draft, Writing – review and editing | Micareme Tchoupo, Methodology | Fleuriane Metissa Dondji Kamga, Methodology | Leon M. J. Mugenzi, Methodology | Flobert Njiokou, Supervision, Writing – review and editing | Grant L. Hughes, Supervision, Writing – review and editing | Charles S. Wondji, Conceptualization, Funding acquisition, Supervision, Writing – review and editing

## DATA AVAILABILITY

Data set supporting the conclusions of this article is available in GenBank database under the accession numbers PQ321536-PQ321555.

## ADDITIONAL FILES

The following material is available online.

Open Peer Review

**PEER REVIEW HISTORY (review-history.pdf).** An accounting of the reviewer comments and feedback.

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
