## [Reviewer comments · Microbiology Spectrum]

Microbiology Spectrum

Contrasting patterns of *Asaia* association with *Plasmodium falciparum* between field collected *Anopheles gambiae* and *Anopheles coluzzii* from Cameroon

Claudine Grâce Tatsinkou Maffo, Maurice Marcel Sandeu, Micareme Tchoupo, Fleuriane Metissa Dondji Kamga, Leon Mugenzi, Flobert Njiokou, Grant Hughes, and Charles Wondji

Corresponding Author(s): Maurice Marcel Sandeu, Centre for Research in Infectious Diseases

Review Timeline:

Submission Date:	March 4, 2024
Editorial Decision:	April 26, 2024
Revision Received:	July 13, 2024
Editorial Decision:	August 8, 2024
Revision Received:	September 9, 2024
Accepted:	September 13, 2024

Editor: Joel Vega-Rodríguez

Reviewer(s): The reviewers have opted to remain anonymous.

Transaction Report:

DOI: <https://doi.org/10.1128/spectrum.00567-24>

Re: Spectrum00567-24 (Contrasting patterns of *Asaia* sp association with *Plasmodium falciparum* between field collected *Anopheles gambiae* and *Anopheles coluzzii* from Cameroon)

Dear Dr. Maurice Marcel Sandeu:

Thank you for the privilege of reviewing your work. Below you will find my comments, instructions from the Spectrum editorial office, and the reviewer comments.

Please make sure to proofread the manuscript by a native speaker to enhance readability.

Revision Guidelines

Sincerely,
Joel Vega-Rodríguez
Editor
Microbiology Spectrum

Reviewer #1 (Comments for the Author):

General comments:

This is a generally useful study of the distribution of *Asaia* in two ecologically distinct regions in Cameroon in a geographical area where there is ongoing *Plasmodium* transmission. As the authors point out, *Asaia* may become a useful paratransgenesis

platform for use in the field so this study is relevant for that. In a sense, these data are the field version of Capone et al. 2013 (the author's reference 16) but without the microscopy. Capone found that *Asaia* and *Plasmodium* basically don't affect each other in the mosquito. And is what the current authors found in the field. I think that is useful, but not necessarily earth-shaking. However, at several points in the manuscript, the authors want to find some kind of correlation between *Asaia* and *Plasmodium*. Those kinds of speculation need to be removed. I think even these authors know that is true since they more or less say this in the abstract and, eventually, in the discussion.

Factual issues and other queries:

Line 82: *Pantoea* is not known to spread through mosquito populations. You actually state this in line 84!

Line 93. ... *S. fuercifera* is a planthopper and is definitely NOT transmitting human disease. Maybe take it out of the list of arthropods carrying *Asaia* that vector human disease.

Line 94. You need a reference for *Asaia* causing few human diseases. Maybe just cite the very few papers that do show human infections (there are a total of only about 6 in all of Pubmed).

Line 133: What is the temperature at which the cultures were grown?

Line 158: What is the concentration of the DNA in this reaction? Only a volume is given.

Line 222: "Completely similar" to me means identical. If there is no sequence difference then they are identical and it sure looks like many of your isolates in your tree are identical since their branches are all attached to the same branch.

Line 252: What does "all had homology with *Aaia* sp" mean? Were they identical? How similar were they. Please state a value.

Line 258-59: You are suggesting that the variants are locality-specific but your data do not show this!

Lines 273: *Asaia* infects nearly all of the females so, of course, if they are also infected with *Plasmodium* they will co-occur.... This is not a surprise.

Line 329: The references here for "other microbes" are only to *Wolbachia*. You should state the and the more general "other microbes" phrase.

Line 333: The sentence is claiming that *Asaia* has a negative effect on *Plasmodium* at different life stages, but the reference is to a paper NOT using *Asaia*. As far as we know, wild-type *Asaia* don't affect *Plasmodium*. The author's own data seem to show that.....

Line 340: reference 42 is to a paper that shows that transgenic *Asaia* can inhibit *Plasmodium*, not wild-type. Throughout this manuscript wherever *Asaia* is stated to be inhibitory to *Plasmodium*, the authors are usually referencing papers where the *Asaia* has been engineered to be inhibitory. Wild type *Asaia* do not seem to be inhibitory to *Plasmodium* in the sense that *Asaia* secretes some anti-*Plasmodium* factor. The other paper cited here (number 16) only suggests that *Asaia* might be activating the mosquito immune system and that that, in turn, might affect parasite numbers but the effect is slight.

Lines 367-69: The authors speculate that the presence of *Plasmodium* may reduce the load of symbionts in the gut. There is no evidence for this at all (their own data included). A paper they cite repeatedly (Capone et al, ref 16) shows the opposite!

Line 549: Are the sequences 16S sequences? If so, state that.

Line 555: Somebody forgot to add the significance sentence!

Figures and tables:

1. I think it is essential to provide the sequence alignments used in this study. Maybe include them in a supplementary figure but I really wish I could have looked at them. It would have helped me interpret the trees and the haplotype figure.

2. Figure 1B and 3A: I don't think you have enough sequences here for circular trees. Please redraw in another format. Also, you used outgroups which infers that you know where to place the root but I don't see a root. Please provide one. Finally, unless I am missing something, almost all of these sequences are identical to each other. You have dozens of sequences attached to the same branch which tells me they are not different from each other. Again, a sequence alignment would have helped here.

3. Figure 3B: the network analysis is confusing and therefore, not convincing. Either describe it better or remove it entirely. Moreover, the haplotypes are not described at all anywhere else. Perhaps use the sequence alignment figure I suggested above to identify the haplotypes you identified.
4. Figure 4: I had a hard time telling whether panels A and B are showing the different collection sites or different Anopheles species. Please add these labels to the figure.
5. Figure 5: This figure needs some color. I would suggest coloring the individual dots for mosquito samples and leaving the average bars and whiskers in black. As it is now, I can't see those because they are washed out in the cloud of black dots. Also, "Amount of Asaia" sounds pretty vague. I would change that label to whatever it is you are actually measuring.

Grammar, English, and style corrections

This paper needs to be reviewed by a native English speaker. There is one listed in the author list so maybe he should have a close look at it since it does not appear that he has done so to date. At a minimum, I suggest the following corrections to the grammar and English in this paper:

Line 30: ...extracted from whole individual....

Line 31: ...qPCR by amplification of the 16S...

Line 32 ...infection in the same mosquitoes.

Line 54 ...qPCR

Line 55 ...infection in the same mosquitoes....

Line 64 ...Africa shoulders ...

Lines 65-66.deaths, with 78.9% of deaths in children under 5 years of age.

Line 70 .. studies have shown

Line 74 ... either suppress population size.... or replace the population with ...

Line 78..One strategy, termed paratransgenesis, exploits symbiotic bacteria, and ...

Line 79 .. fighting malaria mainly by...

Line 93 ...many of which can vector....

Line 96. ...achieving paratransgenesis against...

Line 106.Plasmodium....

Line 109. ...sites and mosquito....

Line 124. Live Anopheles

Line 124. Please decide on a style for "sl" and "ss". Every possible combination of italicized/unitalicized and with and without periods is used in this manuscript. Go through the whole thing and make sure this is consistent. It is my understanding that the correct abbreviation for sensu lato and sensu strictu are "s.l." and "s.s.", respectively. Maybe ask the journal for clarification.

Line 126. ... of Asaia then proceeded...

Line 128. .. and ground in...

Line 131. ..cycloheximide with the pH adjusted....

Line 132. .. ml of enrichment...

Line 134. ...After that, the suspension...

Line 135 ...spread on a CaCO₃....

Line 146. ...and bacterial isolates...

Line 171 ...samples, the relative ratio....

Line 200-01 ...of all positive specimens...

Line 201. ...curve of the 3D7 strain...

Line 208 ...GraphPad Prism.... (also provide a reference!)

Line 209....Chi-squared..... (or X².... Decide on one and use it throughout)

Line 216... s.l.

Line 217colonies differing in color.

Line 222 ...identical and located in...

Line 228.... s.s.

Line 237. ...infection, and strain diversity....

Line 239. s.s.

Line 250. In addition, we investigated...

Line 277.=0.0988), ...

Line 296. ...within the insect vector (ref?)....

Line 314.season and locality.

Line 319 A previous study....

Line 321. comparisons of

Line 325. ...host age and physiological....

Line 326. ...density which is observed...

Line 328. ...microbiome and affect the host...

Line 335. ...Asaia and Plasmodium.....gambiae s.l. and coluzzii populations....

Line 343. In contrast, ...

Line 349. Bai et al.

Line 353. This indicates that the

Line 355. ...also suggested that, despite the fact that there is overlap...

Line 359. Capone et al.

Line 530. ...on glucose-yease extract....

Line 533. ...(c) and bright pink....

Line 535. A phylogenetic tree....

Line 536. The maximum likelihood....

Line 537 ...was used to infer evolutionary relationships and associated taxa....

Reviewer #2 (Comments for the Author):

In this work Maffo, Sandeu and co-authors describe the prevalence and relative abundance of *Asaia* bacteria in *Anopheles* sp field mosquitoes in two different localities in Cameroon. They show that *Asaia* loads differ between *An. gambiae* and *An. coluzzii* mosquitoes, and between mosquitoes collected during the dry and rainy season. The authors found some differences in *Asaia* loads according to *Plasmodium* infectious status, but the trend was opposite between *An. gambiae* and *coluzzii* mosquitoes. No overall correlation was found between *Plasmodium* and *Asaia* prevalence or relative abundances. The study is well designed, and the results are particularly interesting in the context of *Asaia* paratransgenesis applications. I have some suggestion that would increase the clarity of the paper.

Major comments:

- The authors should discuss/disclaim that *Plasmodium* infection status was determined in mosquitoes only by PCR, therefore it is not possible to draw any conclusion on the direct correlation between *Asaia* presence/load and *Plasmodium* development in the mosquito. For instance, if mosquitoes would just have ingested *Plasmodium*-containing blood before being captured, *Plasmodium* DNA could be detected by PCR but this would not mean that the mosquito is infected by *Plasmodium*. Some parts of the text should be changed accordingly (i.e. lines 41 and 58, "is not correlated to *P. falciparum* development" -> change it to "presence").
- The primers used to quantify *Asaia* have been designed in this study, but there is no description on the design procedure and Table S1 with primer sequences is missing. Missing information should be added.
- The mosquito numbers in Table 1, 2 and 3 should be checked: for instance, in the total number of mosquitoes collected in Gounougou are $265 + 159 = 424$ (Table 2) or $60 + 364 = 424$ (Table 3), while it should be 425 according to Table 1. Moreover, in the corresponding text (lines 262-278) numbers/percentages are different from those reported in the Tables: for example at line 264 the number of *An. gambiae* mosquitoes co-infected with *Plasmodium* and *Asaia* is 58/488 while the total number of mosquitoes is 439 according to Table 3.
- The total number of mosquitoes analysed is different at each step (identification of mosquito species, *Plasmodium* infection status, *Asaia* loads) and it is a bit confusing. A summary chart showing the subset of mosquitoes analysed per site/species would help to clarify. Moreover, it is quite confusing that mosquito species also corresponds to the sampling site. Titles to figures 2-4-5-6 could help clarifying (for example Figure 2B could mention "*An. gambiae* - Bankeng" and Figure 2C "*An. coluzzii* - Gounougou").
- Statistical tests used for each analysis should be reported either in the text or in the figure legend.

Minor comments:

- Please, be consistent in the terminology: real-time Q-PCR (lines 31 and 54) vs qPCR (line 102 or line 156) vs Real-time PCR (line 153) vs quantitative PCR (line 238).
- "16S ribosomal RNA" or "16rRNA" should be italicized and consistency should be used throughout the text.
- References 3-6 are referred to Cameroon, while the text is a general introduction on malaria and *Anopheles* mosquitoes in Africa. I suggest to cite other papers that describe similar phenomena in other African countries.
- Line 78: missing space between "has" and "been".
- Line 94: missing reference to the affirmation that bacteria of the genus *Asaia* cause few human infections.
- Line 132-133: it is not clear if the mentioned handmade enrichment culture medium is the same medium mentioned at lines 129-131.
- Line 159: the full stop at the end of the line should be removed.
- Lines 200-202: please, rephrase as this sentence is not clear.
- Statistics should be described in a separate paragraph.
- Lines 228-229: a verb is missing.
- Line 245: please correct the median value.
- The legend of Figure 4 is incomplete I think ("Add significance sentence").
- Reference 33 = reference 27.
- In Figure 2 and 5 scale bars are not always visible: another colour could be used to help visualization.

MICROBIOLOGY SPECTRUM

RE: SPECTRUM00567-24

Dear, Joel Vega-Rodríguez,

We are pleased that you find the manuscript of interest and that you are willing to consider a revised version addressing the concerns raised by the reviewers of the manuscript.

We have considered all the comments and suggestions made by the reviewers, as well as the editorial comments made by your office. We hereby submit a revised version of the manuscript (all changes highlighted in red), in the hope that you will find it suitable for publication in Microbiology spectrum.

Please find below our response to the editor-in-chief and reviewer's comments:

Editor-in-Chief:

Reply: Thanks to the Editor-in-Chief for this comment which allows us to improve the quality of the manuscript. We have formatted the new version of the manuscript with the English editing and figures with TIFF files (300-600 dpi, with LZW). You can see all these changes on the revised manuscript with tracked changes.

Reviewer #1:

General comments:

This is a generally useful study of the distribution of *Asaia* in two ecologically distinct regions in Cameroon in a geographical area where there is ongoing Plasmodium transmission. As the authors point out, *Asaia* may become a useful paratransgenesis platform for use in the field, so this study is relevant for that. In a sense, these data are the field version of Capone et al. 2013 (the author's reference 16) but without the microscopy. Capone found that *Asaia* and Plasmodium basically don't affect each other in the mosquito. And is what the current authors found in the field. I think that is useful, but not necessarily earth-shaking. However, at several points in the manuscript, the authors want to find some kind of correlation between *Asaia* and Plasmodium. Those kinds of speculation need to be removed. I think even these authors know that is true since they more or less say this in the abstract and, eventually, in the discussion.

Reply: Thanks to the reviewer for this comment which allow us to improve the quality of the paper. However, we really stated that there is no correlation between *Asaia* and *Plasmodium* in the manuscript. This has been mentioned in the abstract as follow: "However, no correlation was observed between the amount of *Asaia* and *P. falciparum* infection. This study provides evidence that naturally occurring *Asaia* infection is not correlated to *P. falciparum* development within *An. gambiae* and *An. coluzzii*" (Line 39-41).

Factual issues and other queries:

Line 82: *Pantoea* is not known to spread through mosquito populations. You actually state this in line 84!

Reply: Thanks to the reviewer for this remark. I agreed with you, this sentence has been removed.

Line 93. .. *S. fuercifera* is a planthopper and is definitely NOT transmitting human disease. Maybe take it out of the list of arthropods carrying *Asaia* that vector human disease.

Reply: We have updated this section as requested by the reviewer, the *S. fuercifera* has been removed in the manuscript

Line 94. You need a reference for *Asaia* causing few human diseases. Maybe just cite the very few papers that do show human infections (there are a total of only about 6 in all of Pubmed).

Reply: We agree with the reviewer, the reference for *Asaia* causing few human diseases have been added in the manuscript. For, example we have: Snyder, R.W., et al. 2004, Tuuminen et al. 2006, Juretschko et al. 2010, Carretto et al. 2016

Line 133: What is the temperature at which the cultures were grown?

Reply: We agree with the reviewer, the temperature at which the cultures were grown was at 30°C. This has been included in the manuscript.

Line 158: What is the concentration of the DNA in this reaction? Only a volume is given.

Reply: Thank you so much for this question. So, after DNA extraction, we had to select randomly some sample and measure the DNA concentration, all the samples had a good quality DNA with a ratio of 1.7–2.0 at A_{260}/A_{280} . We didn't measure the concentration of DNA for all the samples

Line 222: "Completely similar" to me means identical. If there is no sequence difference, then they are

identical and it sure looks like many of your isolates in your tree are identical since their branches are all attached to the same branch.

Reply: We agree with the reviewer, similar has been replace by identical in the manuscript

Line 252: What does "all had homology with *Aasia* sp" mean? Were they identical? How similar were they. Please state a value.

Reply: We agree with the reviewer, homology has been replaced by identity in the manuscript and the value of the homology with *Asaia* sp was ranging from 99% to 100%.

Line 258-259: You are suggesting that the variants are locality-specific, but your data do not show this!

Reply: Thank so much for this comment, so the variants are localities dependent due to the fact that the haplotype network have to show eight distinct nucleotide sequence variants (Fig. 3B) where the major one H1 and the others H2, H3, H3, H4 and H5 are present only at Bankeng. In addition, the nucleotide sequence variants H6, H7 and H8 are present only in Gounougou.

Lines 273: *Asaia* infects nearly all of the females so, of course, if they are also infected with *Plasmodium*, they will co-occur.... This is not a surprise.

Reply: We agree with the reviewer

Line 329: The references here for "other microbes" are only to *Wolbachia*. You should state the and the more general "other microbes" phrase.

Reply: We agree with the reviewer, and we have added *Wolbachia* to be more precise.

Line 333: The sentence is claiming that Asaia has a negative effect on Plasmodium at different life stages, but the reference is to a paper NOT using Asaia. As far as we know, wild-type Asaia don't affect Plasmodium. The author's own data seem to show that.....

Reply: We agree with the reviewer, this reference has been changed by another one: Bongio et al.2015.

Line 340: reference 42 is to a paper that shows that transgenic Asaia can inhibit Plasmodium, not wild-type. Throughout this manuscript wherever Asaia is stated to be inhibitory to Plasmodium, the authors are usually referencing papers where the Asaia has been engineered to be inhibitory. Wild type Asaia do not seem to be inhibitory to Plasmodium in the sense that Asaia secretes some anti-Plasmodium factor. The other paper cited here (number 16) only suggests that Asaia might be activating the mosquito immune system and that that, in turn, might affect parasite numbers but the effect is slight.

Reply: Thank so much for this comment, so based on the fact that the mechanism underlying on the inhibition of the Plasmodium by wild type Asaia is not yet know, we have just speculated on different mechanisms such Asaia engineered and mosquitos' immune system that Asaia could be used to inhibe Plasmodium infection in mosquitoes.

Lines 367-69: The authors speculate that the presence of Plasmodium may reduce the load of symbionts in the gut. There is no evidence for this at all (their own data included). A paper they cite repeatedly (Capone et al, ref 16) shows the opposite!

Reply: Thanks for this comment. The reference has been changed to the right one (Sharma et al. 2020)

Line 549: Are the sequences 16S sequences? If so, state that.

Reply: We agree with the reviewer, the 16S sequences has been added.

Line 555: Somebody forgot to add the significance sentence!

Reply: Thanks for this comment. It was a mistake; the sentence has been removed

Figures and tables:

1. I think it is essential to provide the sequence alignments used in this study. Maybe include them in a supplementary figure but I really wish I could have looked at them. It would have helped me interpret the trees and the haplotype figure.

Reply: The sequence alignments have been included in the submission as a miscellaneous file

2. Figure 1B and 3A: I don't think you have enough sequences here for circular trees. Please redraw in another format. Also, you used outgroups which infers that you know where to place the root but I don't see a root. Please provide one. Finally, unless I am missing something, almost all of these sequences are identical to each other. You have dozens of sequences attached to the same branch which tells me they are not different from each other. Again, a sequence alignment would have helped here.

Reply: Thanks for this comment. We don't have too many sequences, but it should be better to draw it in circular because it took less place and easy to present on a page rather than traditional/rectangular form

3. Figure 3B: the network analysis is confusing and therefore, not convincing. Either describe it better or remove it entirely. Moreover, the haplotypes are not described at all anywhere else. Perhaps use the sequence alignment figure I suggested above to identify the haplotypes you identified.

Reply: Thanks for this comment. The haplotype network has been redone and the haplotypes have been described according to the samples and localities. The sequence alignment is already provided in the supplementary file.

4. Figure 4: I had a hard time telling whether panels A and B are showing the different collection sites or different *Anopheles* species. Please add these labels to the figure.

Reply: Thank for this suggestion, these labels have been added to the figure.

5. Figure 5: This figure needs some color. I would suggest coloring the individual dots for mosquito samples and leaving the average bars and whiskers in black. As it is now, I can't see those because they are washed out in the cloud of black dots. Also, "Amount of *Asaia*" sounds pretty vague. I would change that label to whatever it is you are actually measuring.

Reply: Thanks for this comment. The individual dots for mosquito samples have been colored. Actually, we are quantifying the bacteria *Asaia* in samples that's why "Amount of *Asaia*" has been used. However, it has been changed to "*Asaia* load".

Grammar, English, and style corrections

This paper needs to be reviewed by a native English speaker. There is one listed in the author list so maybe he should have a close look at it since it does not appear that he has done so to date. At a minimum, I suggest the following corrections to the grammar and English in this paper:

Reply: We thank the reviewer for these suggestions. The manuscript has been greatly improved by a two native English speaker.

Reviewer #2 (Comments for the Author):

In this work Maffo, Sandeu and co-authors describe the prevalence and relative abundance of *Asaia* bacteria in *Anopheles* sp field mosquitoes in two different localities in Cameroon. They show that *Asaia* loads differ between *An. gambiae* and *An. coluzzii* mosquitoes, and between mosquitoes collected during the dry and rainy season. The authors found some differences in *Asaia* loads according to *Plasmodium* infectious status, but the trend was opposite between *An. gambiae* and *coluzzii* mosquitoes. No overall correlation was found between *Plasmodium* and *Asaia* prevalence or relative abundances. The study is well designed, and the results are particularly interesting in the context of *Asaia* paratransgenesis applications. I have some suggestion that would increase the clarity of the paper.

Major comments:

- The authors should discuss/disclaim that *Plasmodium* infection status was determined in mosquitoes only by PCR, therefore it is not possible to draw any conclusion on the direct correlation between *Asaia* presence/load and *Plasmodium* development in the mosquito. For instance, if mosquitoes would just have ingested *Plasmodium*-containing blood before being captured, *Plasmodium* DNA could be detected by PCR but this would not mean that the mosquito is infected by *Plasmodium*. Some parts of the text should be changed accordingly (i.e. lines 41 and 58, "is not correlated to *P. falciparum* development" -> change it to "presence").

Reply: Thanks to the reviewer for this comment, now we have clearly stated in the manuscript that the collected mosquitoes were kept in individual tubes for 7 days post collection, which is the necessary time for *Plasmodium* to develop in the *Anopheles*

mosquito. That's why, we stated *P. falciparum* development in the manuscript. This sentence has been added in the manuscript (Line 121-123).

- The primers used to quantify *Asaia* have been designed in this study, but there is no description on the design procedure and Table S1 with primer sequences is missing. Missing information should be added.

Reply: Thanks for the comment. The primers have been designed using Primer 3 tool. This has been added in the text. In addition, the primers used to quantify *Asaia* has been added in the manuscript.

- The mosquito numbers in Table 1, 2 and 3 should be checked: for instance, in the total number of mosquitoes collected in Gounougou are $265 + 159 = 424$ (Table 2) or $60 + 364 = 424$ (Table 3), while it should be 425 according to Table 1. Moreover, in the corresponding text (lines 262-278) numbers/percentages are different from those reported in the Tables: for example, at line 264 the number of *An. gambiae* mosquitoes co-infected with *Plasmodium* and *Asaia* is 58/488 while the total number of mosquitoes is 439 according to Table 3.

Reply: Thanks for this remark. The number of mosquitoes has been checked and modified. For the line 264, you are right, it was a mistake. It is 58/388, 388 which represents the total of *Asaia* positive mosquitoes at Bankeng rather than 58/488. For the mosquitoes collected at Gounougou, the total number is 424, this has been changed in table 1. The number of mosquitoes for every experiment is the same in all the localities. The mosquito's species is the same as sampling site just because the 2 localities do not have the same species.

- The total number of mosquitoes analysed is different at each step (identification of mosquito species, *Plasmodium* infection status, *Asaia* loads) and it is a bit confusing. A summary chart showing the subset of mosquitoes analysed per site/species would help to clarify. Moreover, it is quite confusing that mosquito species also corresponds to the sampling site. Titles to figures 2-4-5-6 could help clarifying (for example Figure 2B could mention "*An. gambiae* - Bankeng" and Figure 2C "*An. coluzzii* - Gounougou").

Reply: Thank for this suggestion, these labels ("*An. gambiae* - Bankeng"; "*An. coluzzii* - Gounougou") have been added (to the figure).

- Statistical tests used for each analysis should be reported either in the text or in the figure legend.

Reply: Thanks for this comment. The statistical tests used for each analysis have been reported in the section "statistical analysis" in the text.

Minor comments:

- Please, be consistent in the terminology: real-time Q-PCR (lines 31 and 54) vs qPCR (line 102 or line 156) vs Real-time PCR (line 153) vs quantitative PCR (line 238).

Reply: We have updated this section as requested by the reviewer and this has been harmonized by qPCR in all the manuscript

- "16S ribosomal RNA" or "16rRNA" should be italicized and consistency should be used throughout the text.

Reply: This has been corrected and harmonized by *16S ribosomal RNA* in all the manuscript

- References 3-6 are referred to Cameroon, while the text is a general introduction on malaria and *Anopheles* mosquitoes in Africa. I suggest to cite other papers that describe similar phenomena in other African countries.

Reply: We have updated this section as requested by the reviewer by adding new reference in *Anopheles* mosquitoes in Africa (references 7-9).

- Line 78: missing space between "has" and "been".

Reply: This has been corrected

- Line 94: missing reference to the affirmation that bacteria of the genus *Asaia* cause few human infections.

Reply: Thanks for your comment. References for *Asaia* causing few human diseases have been cited –(Snyder, R.W., et al. 2004, Tuuminen et al. 2006, Juretschko et al. 2010, Carretto et al. 2016)

- Line 132-133: it is not clear if the mentioned handmade enrichment culture medium is the same medium mentioned at lines 129-131.

Reply: It is not the same medium and the composition of each of them has been clearly mentioned in the manuscript.

- Line 159: the full stop at the end of the line should be removed.

Reply: We agree with the reviewer. The full stop at the end of the line has been removed

- Lines 200-202: please, rephrase as this sentence is not clear.

Reply: This sentence has been rephrase as follow: Absolute quantification of all positive specimens was done using the standard curve of the 3D7 strain DNA and generated from serial dilution methods resulting in a quantification range of 7.10¹ to 7. 10⁵ genomes/ μ L

- Statistics should be described in a separate paragraph.

Reply: This has been corrected

- Lines 228-229: a verb is missing.

Reply: This has been corrected

- Line 245: please correct the median value.

Reply: This has been corrected

- The legend of Figure 4 is incomplete I think ("Add significance sentence").

Reply: Thanks for this comment. It was a mistake; the sentence has been removed.

- Reference 33 = reference 27.

Reply: This has been corrected

- In Figure 2 and 5 scale bars are not always visible: another colour could be used to help visualization.

Reply: Thanks for this comment. The individual dots for mosquito samples have been colored. Actually, we are quantifying the bacteria *Asaia* in samples that's why "Amount of *Asaia*" has been used.

Re: Spectrum00567-24R1 (Contrasting patterns of *Asaia* association with *Plasmodium falciparum* between field collected *Anopheles gambiae* and *Anopheles coluzzii* from Cameroon)

Dear Dr. Maurice Marcel Sandeu:

Thank you for the privilege of reviewing your work. Below you will find my comments, instructions from the Spectrum editorial office, and the reviewer comments.

Revision Guidelines

Sincerely,
Joel Vega-Rodríguez
Editor
Microbiology Spectrum

Reviewer #1 (Comments for the Author):

Most of the comments I had from the initial submission of this manuscript have been addressed. Please address the following:

Major comments

1. Figure 1 A. All of the Asaia strains identified in this figure from this work fall into the same clade as the published sequences. In fact, the tree indicates they are identical and form a 32-branch polytomy. There is no resolution indicating species, only genus. Please rewrite lines 224-25 to reflect this.
2. Fig 3A. The authors state that there is clustering of Asaia strains in both localities. That's not what the tree shows. Again, this tree has a gigantic polytomy right after the implied root leading to essentially no resolution between the samples at all. There are a couple minor clades (also within the polytomy) but that's it. Coloring some branches in a polytomy and calling that "clustering" is not valid.
3. Sequence alignment: Is the sequence alignment provided intended to be a figure? Even a supplemental one? It is not formatted as one. Also, it is impossible to see any sequence differences when they are presented like this. Please change the alignment so that the entire first sequence is presented, and then only changes from that sequence in every other row.
4. As the other reviewer noted in the first round of review, the statistical tests need to be named and the parameters given in the figure legends where they are used (figures 2, 5, and 6). These are still lacking.

Minor comments

1. The formatting of "s.l." and "s.s." is still inconsistent. Please check throughout and use a single style.
2. Line 61: Please use the 2022 WHO data which is the most recent.
3. Lines 187-188: What does "clarity of the sequences" mean?
4. Figure 1 legend: please explain the coloring of the branches in the tree.

Style comments

I suggest the following changes in language in each numbered line below:

Line 25: "..makes it a potential candidate.."

Line 42: "Further studies incorporating experimental infections are required.."

Lines 80-81: "For example, *Serratia marcescens* AS1 can propagate through mosquito populations while *Pantoea agglomerans* can significantly reduce *P. falciparum* development when engineered..."

Lines 134-35: "...10.0 mls of enrichment culture...."

Line 183: "...using the 16S..."

Line 184: "...included 19 and 20..."

Line 189: "...samples were purified using..."

Line 215: "...using the Chi-squared..."

Lines 266-67: "...not entirely locality dependent."

Line 301: "..in the mosquito populations we studied."

Reviewer #2 (Comments for the Author):

Most of my comments were taken into account and I find this version of the manuscript much improved.

I only have few minor comments:

- 1) There are still some discrepancies in numbers found in the text and Table 3. Please check carefully. For instance, the total number of mosquitoes analyzed from Bankeng should be 439, not 388 (L. 272). In Gounougou, is the number of mosquitoes co-infected with Asaia and Plasmodium 60/423 or 59/423 (L. 273)?
- 2) Error bars in Figure 2 are really small compared to Figure 5, although data in Figure 5 is a subset of data in Figure 2 and both figures represent median with 95% CI. Could the authors check if there is any mistake?
- 3) Information on the statistical test used to correlate Asaia and Plasmodium loads is missing in the corresponding part in materials and methods, and the information can't be found in the main text or in the figure legend.

Typos:

L. 121: "vetors"

L. 134: "10 ml of handmade ml"

L. 137: CaCO3 -> CaCO₃

L: 158: 16s -> 16S

L. 185: ...from Bankeng (missing "collected") during the dry and rainy season...

L. 186: ...from Gounougou (missing "collected") during the dry and rainy season...

L. 278: Final "a" is missing from Asaia

- L. 351: "non" is in italics.
- L. 360-361: check this sentence as it sounds weird.
- L. 362: "that" is repeated twice.
- L. 395: remove the ";

MICROBIOLOGY SPECTRUM

RE: SPECTRUM00567-24

Dear, Joel Vega-Rodríguez,

We are pleased that you find the manuscript of interest and are willing to consider a revised version that addresses the concerns raised by the reviewers.

We have carefully considered all the comments and suggestions provided. We hereby submit a revised version of the manuscript, with all changes highlighted in red, in the hope that you will find it suitable for publication in *Microbiology Spectrum*.

Please find below our response to the reviewer's comments:

Reviewer #1 (Comments for the Author):

Most of the comments I had from the initial submission of this manuscript have been addressed. Please address the following:

Major comments

1. Figure 1 A. All of the *Asaia* strains identified in this figure from this work fall into the same clade as the published sequences. In fact, the tree indicates they are identical and form a 32-branch polytomy. There is no resolution indicating species, only genus. Please rewrite lines 224-25 to reflect this.

Reply: Thanks for this comment. Indeed, there is no species-level resolution because the tools used here do not allow for identification down to the species level. The lines 224-25 have been revised accordingly.

2. Fig 3A. The authors state that there is clustering of *Asaia* strains in both localities. That's not what the tree shows. Again, this tree has a gigantic polytomy right after the implied root leading to essentially no resolution between the samples at all. There are a couple minor clades (also within the polytomy) but that's it. Coloring some branches in a polytomy and calling that "clustering" is not valid.

Reply: Thank you for the comment. There were actually just minor clades with low bootstrap values. The term "clustering" was used to indicate that the Asaia strains identified in this work and highlighted in colour fall within the same major clade.

3. Sequence alignment: Is the sequence alignment provided intended to be a figure? Even a supplemental one? It is not formatted as one. Also, it is impossible to see any sequence differences when they are presented like this. Please change the alignment so that the entire first sequence is presented, and then only changes from that sequence in every other row.

Reply: The alignment has been adjusted so that the entire first sequence is presented, with subsequent rows showing only the changes relative to that sequence.

4. As the other reviewer noted in the first round of review, the statistical tests need to be named and the parameters given in the figure legends where they are used (figures 2, 5, and 6). These are still lacking.

Reply: Thanks for the comments. This has been added.

Minor comments

1. The formatting of "s.l." and "s.s." is still inconsistent. Please check throughout and use a single style.

Reply: This has been corrected

2. Line 61: Please use the 2022 WHO data which is the most recent.

Reply: We agree with the reviewer. This has been changed

3. Lines 187-188: What does "clarity of the sequences" mean?

Reply: Thanks for the comment. This has been changed to "sharpness of the bands after the gel electrophoresis"

4. Figure 1 legend: please explain the coloring of the branches in the tree.

Reply: Thanks for the comment. This change has been made in the Figure 1 legend.

Style comments

I suggest the following changes in language in each numbered line below:

Line 25: "...makes it a potential candidate.."

Reply: This has been corrected

Line 42: "Further studies incorporating experimental infections are required."

Reply: This has been corrected

Lines 80-81: "For example, *Serratia marcescens* AS1 can propagate through mosquito populations while *Pantoea agglomerans* can significantly reduce *P. falciparum* development when engineered..."

Reply: This has been corrected

Lines 134-35: "...10.0 mls of enrichment culture...."

Reply: This has been corrected

Line 183: "...using the 16S..."

Reply: This has been corrected

Line 184: "...included 19 and 20..."

Reply: This has been corrected

Line 189: "...samples were purified using..."

Reply: This has been corrected

Line 215: "...using the Chi-squared..."

Reply: This has been corrected

Lines 266-67" "...not entirely locality dependent."

Reply: This has been corrected

Line 301: "...in the mosquito populations we studied."

Reply: This has been corrected

Reviewer #2 (Comments for the Author):

Most of my comments were taken into account and I find this version of the manuscript much improved.

I only have few minor comments:

1) There are still some discrepancies in numbers found in the text and Table 3. Please check carefully. For instance, the total number of mosquitoes analyzed from Bankeng should be 439, not 388 (L. 272). In Gounougou, is the number of mosquitoes co-infected with *Asaia* and *Plasmodium* 60/423 or 59/423 (L. 273)?

Reply: Thank you for the comment. Indeed, 388 represents the number of *Asaia*-positive samples at Bankeng (see Table 1). We did not work with negative samples to compare the amount of *Asaia* between the localities. In Gounougou, the number of mosquitoes co-infected with *Asaia* and *Plasmodium* is 60 out of 423. This has been modified accordingly.

2) Error bars in Figure 2 are really small compared to Figure 5, although data in Figure 5 is a subset of data in Figure 2 and both figures represent median with 95% CI. Could the authors check if there is any mistake?

Reply: Thanks for the comment. The error bars have been harmonized.

3) Information on the statistical test used to correlate *Asaia* and *Plasmodium* loads is missing in the corresponding part in materials and methods, and the information can't be found in the main text or in the figure legend.

Reply: Thanks for the comment. This has been added to the materials and methods section see

Typos:

L. 121: "vetors"

Reply: This has been corrected

L. 134: "10 ml of handmade ml"

Reply: This has been corrected

L. 137: CaCO3 -> CaCO₃

Reply: This has been corrected

L: 158: 16s -> 16S

Reply: This has been corrected by 16S ribosomal RNA

L. 185: ...from Bankeng (missing "collected") during the dry and rainy season...

Reply: This has been corrected

L. 186: ...from Gounougou (missing "collected") during the dry and rainy season...

Reply: This has been corrected

L. 278: Final "a" is missing from Asaia

Reply: This has been corrected

L. 351: "non" is in italics.

Reply: This has been corrected

L. 360-361: check this sentence as it sounds weird.

Reply: This has been corrected

L. 362: "that" is repeated twice.

Reply: This has been corrected

L. 395: remove the ";

Reply: This has been corrected

Re: Spectrum00567-24R2 (Contrasting patterns of *Asaia* association with *Plasmodium falciparum* between field collected *Anopheles gambiae* and *Anopheles coluzzii* from Cameroon)

Dear Dr. Maurice Marcel Sandeu:

Your manuscript has been accepted, and I am forwarding it to the ASM production staff for publication. Your paper will first be checked to make sure all elements meet the technical requirements. ASM staff will contact you if anything needs to be revised before copyediting and production can begin. Otherwise, you will be notified when your proofs are ready to be viewed.

IMPORTANT: The authors need to update the accession numbers for data deposition in Gene Bank during the copy-editing stage before publication.

Sincerely,
Joel Vega-Rodríguez
Editor
Microbiology Spectrum